# Developments and Industrial Applications of Basalt Fibre Reinforced Composite Materials

Indraneel R. Chowdhury [ID], Richard Pemberton [ID] and John Summerscales *[ID]

School of Engineering, Computing and Mathematics, University of Plymouth, Drake Circus, Plymouth PL4 8AA, UK
* Correspondence: j.summerscales@plymouth.ac.uk

**Abstract:** Basalt mineral fibre, made directly from basalt rock, has good mechanical behavior, superior thermal stability, better chemical durability, good moisture resistance and can easily be recycled when compared to E-glass fibres (borosilicate glass is called 'E-glass' or 'electric al-grade glass' because of its high electrical resistance) which are traditionally used in structural composites for industrial applications. Industrial adoption of basalt fibre reinforced composites (FRC) is still very low mainly due to inadequate data and lower production volumes leading to higher cost. These reasons constrain the composites industry from seriously considering basalt as a potential alternative to conventional (e.g., E-glass) fibre reinforced composites for different applications. This paper provides a critical review of the state-of-the-art concerning basalt FRC highlighting the increasing trend in research and publications related to basalt composites. The paper also provides information regarding physico-chemical, and mechanical properties of basalt fibres, some initial Life cycle assessment inventory data is also included, and reviews common industrial applications of basalt fibre composites.

**Keywords:** basalt; mechanical properties; durability; life cycle assessment; industrial applications

## 1. Introduction

FRC materials have a wide spectrum of applications ranging from aerospace components to automotive products, energy, and from construction industry to marine engineering, electronic and chemical packaging, and medical equipment [1–3]. In many applications, FRC materials are developed to replace metal components, particularly those used in corrosive environments. The main advantages of FRC as an alternative to metallic materials are that FRC materials provide a better sustainability, (a balance of technical, economic, environmental, social and governance aspects), for industrial applications as compared to metallic based components [1,4].

In FRCs, reinforcement fibres are available in different fabric architectures patterns (e.g., uniaxial, woven, knitted, non-crimp, multiaxial) with a predefined orientation for the main reinforcing material embedded in a polymeric matrix (typically a thermosetting or thermoplastic one) [5]. Most commonly used fibre reinforcements for FRC in industrial applications are petroleum-derived carbon, glass or aramid fibres [1,4]. Use of FRC in industrial applications can provide a significant advantage over metallic materials as there can be a notable improvement in the structural, mechanical, and tribological properties of FRC industrial products [1–5].

One major concern remains the recycling of thermoset materials at the end-of-life. Different composite recycling processes (e.g., thermal recycling by pyrolysis, chemical recycling (solvolysis) or high voltage fragmentation (HVF)) have been developed to cope with the tons of composite wastes generated every year [5]. Most of these techniques are not cost-effective requiring high heat energy to operate. Retaining mechanical and volumetric integrity of recycled fibres also remains a major issue [6]. Therefore, dumping to landfill remains the most pragmatic way to handle composite wastes [5,6]. Recycling of

the fibres remains a key challenge. Recycling techniques like thermal incineration, are not suitable due to lack of thermal stability of these fibres at elevated temperatures, leading to a serious reduction in the strength of the recycled fibres. Landfilling of composite wastes manufactured using fibre reinforcements has anthropogenic environmental effects.

Over recent decades, considering the environmental issues and challenges associated with composite recycling, there has been an increasing trend towards developing sustainable composites for industrial applications using various natural fibres (e.g., different vegetable fibres such as hemp, flax, jute, kenaf, wood and silk or wool animal fibres) as reinforcement [7–10]. Most natural fibres are hydrophilic, and the reinforced composites tend to have poor performance relative to the traditional synthetic fibre reinforced composites [10,11]. One potential alternative can be basalt, which is a mineral fibre derived directly from volcanic basalt rocks without secondary additives. Basalt fibres have better physico-chemical and mechanical properties than synthetic fibres (such as E-glass), have good thermal stability at high temperatures, good moisture and chemical resistance, good sound and heat insulation properties and good processability. Basalt is unlikely to compete with carbon fibres (CF) given the much higher elastic modulus of CF. The manufacturing process for basalt is significantly cheaper than glass fibres, requiring less energy and no secondary additives during the process. Moreover, basalt is non-toxic in either air or water, non-combustible and explosion-proof, whilst also being eco-friendly [7–9]. Life Cycle Assessment (LCA) results in the literature confirmed that basalt fibre reinforced polymers (FRP) in the construction sector were more environmentally friendly than conventional steel rebars [10,11]. Basalt FRC can provide a good cost-to-quality ratio and have a wide spectrum of industrial applications ranging from the aerospace, automotive, marine and rail industry to the energy, construction, chemical and electrical sectors. Relative cost and a lack of available data still prevents the industry from seriously considering basalt fibres and their composites as a potential alternative to traditional synthetic FRC in various applications [12].

This paper aims to provide a critical review of the current state-of-the-art on basalt FRC materials concerning the data available in open literature with a focus towards highlighting the increasing trend in research and publications related to basalt composites. The paper also covers physico-chemical and mechanical properties of basalt fibres. Finally, the paper highlights common industrial applications of basalt FRC materials and developments made in the field related to physical, mechanical, and chemical characteristics of the material.

## 2. Basalt

Basalt is the most abundant among all igneous rocks, constituting more than 90% of all volcanic rocks. The microstructural constituents of basalt rock are strongly dependent on the rate of cooling of molten lava. When the solidification rate is slow, basalt microstructure demonstrates a potentially crystalline atomic arrangement, while a faster solidification rate leads to an amorphous structure. Mineralogically, basalt mainly consists of three silicate minerals: plagioclases, pyroxenes, and olivines [7–9]. Silicon dioxide ($SiO_2$) is the main chemical component of basalt. Other metal oxides like $Al_2O_3$, $Fe_2O_3$, CaO, MgO, $Na_2O$, $K_2O$, $TiO_2$ also constitute the chemical composition of basalt fibres. The percentage of different metal oxides in basalt fibres, strongly depends on the geographical location of the basalt rocks. Basalt rocks abundantly found all over the world are generally classified according to the $SiO_2$ content and can be categorised as: alkaline (with $SiO_2$ content of ~42%), mildly acidic (with $SiO_2$ content between 43–46%) and acidic basalts (with over 46% $SiO_2$ content) [7,8]. Only acidic basalts are considered suitable for basalt fibre production due to higher $SiO_2$ content, which helps in providing good mechanical and chemical stability to basalt fibres [13,14]. The presence of other metal oxides like $Al_2O_3$ helps in providing good chemical stability. The presence of CaO, MgO and $TiO_2$ provides good moisture and corrosion resistance, while $Fe_2O_3$ confers good thermal stability to basalt fibres [14].

Basalt has been used for many years in casting processes including manufacturing slabs and tiles for architectural applications. Basalt fibres were first extruded from molten basalt rocks in the early 1920s attributed to the work of Paul Dhè [15]. Later in the 1960s, the Soviet Union while investigating the applications of basalt for military and aerospace equipment, also started to develop manufacturing techniques to produce continuous basalt fibres [7]. In 1979, a patent was granted in the United States where researchers proposed a production technology to improve the mechanical, chemical, and physical characteristics of basalt fibres by optimizing the fibre sizing with the addition of silane coupling agents and hydrate zirconia films [16–20]. In 1985, scientists in the Soviet Union developed a commercial production technology to manufacture continuous basalt fibres [21–23]. Other studies focused on developing manufacturing methods and techniques for producing basalt fibres. Aslanova [24] proposed a production technique to improve the mechanical properties and thermal stability of basalt. Brik [25,26] proposed a flexible manufacturing technique and apparatus to produce continuous basalt fibres from 7 to 100 μm diameter.

The global basalt fibre market size is projected to grow from 286 million USD to 517 million USD by 2027, at a Compound annual growth rate (CAGR) of 12.5% between 2022 and 2027 [27]. Current production of basalt fibres is relatively widespread with most of the production market dominated by Ukraine, and Russia, due to large basalt reserves in the respective regions. New basalt fibre manufacturers are rapidly emerging in other countries, such as China, Japan, Ireland, Germany, United states of America (USA), Belgium, to meet the growing demand [27]. To name a few notable manufacturers of basalt fibres globally include—Zhejiang GBF Basalt Fibre (China), JFE RockFibre Corp. (Japan), Mafic SA (Ireland), ISOMATEX SA (Belgium), Kamenny Vek (Russia), Technobasalt-Invest LLC (Ukraine), INCOTELOGY GmBH (Germany), Sundaglass Basalt Fibre Technology (USA), Shanxi Basalt Fibre Technology Co. Ltd. (China), Mudanjiang Basalt Fibre Co. (China) [27]. The main consumers span across North America, Asia, and Europe.

### 2.1. Manufacturing Techniques for Basalt Fibre Production

The major challenges associated with manufacturing basalt fibres are gradual crystallization of structural parts (plagioclase, magnetite, pyroxene) and the non-homogeneity of melt [28]. Continuous spinning technologies can help in overcoming the challenges associated with manufacturing basalt fibres suitable to produce basalt fabrics for structural composites [28,29]. The two main manufacturing techniques for basalt fibres are Spinneret technology [14,30] and Junkers technology [30,31].

The Spinneret technology, used to manufacture continuous basalt fibres, (Figure 1) is similar to the manufacturing technique used for glass fibre production. However, in basalt fibre production, a single raw material feed process requires no secondary additives in the intermediate stages (unlike glass fibre manufacturing process). For basalt fibre production, basalt rock is first crushed and then washed. Afterwards, the broken rock fragments are loaded into the furnace where they are heated to between 1200–1500 °C using either a gaseous mixture or electricity. To ensure uniform heating and to achieve thermal equilibrium in the shortest possible time, some electrodes are submerged in the melting bath. Following the melting process, molten basalt is passed through platinum-rhodium heated bushings to produce fine threads of basalt fibres. After cooling is complete, basalt fibre filaments are collected in strands and further lubricated to retain the integrity and chemical stability of fibres. During the manufacturing process, the diameter of basalt fibres is controlled by varying the drawing speed and melting temperature [7,9,14,30].

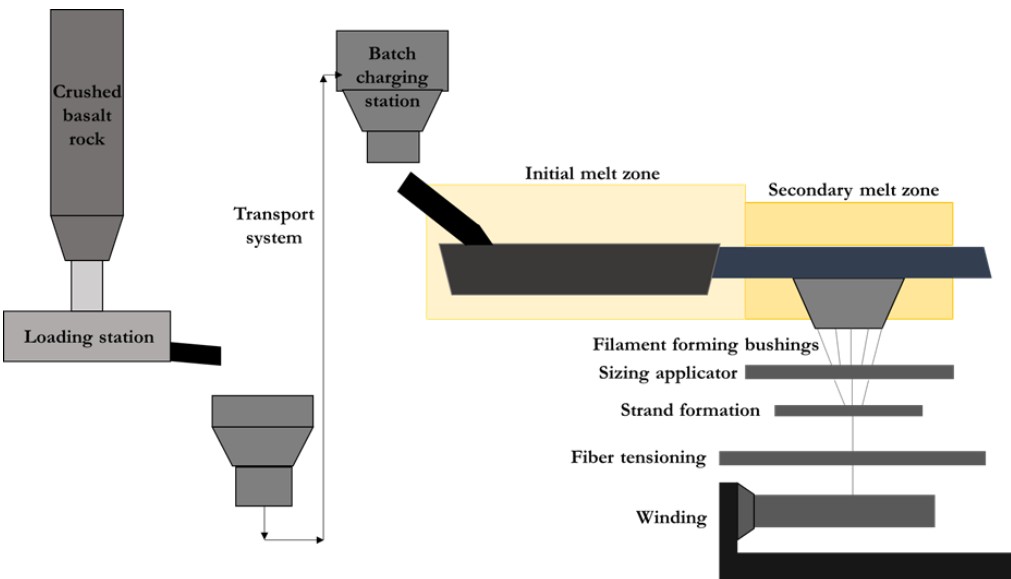

**Figure 1.** Schematic of Spinneret technology (redrawn from [7]).

Junkers technology (Figure 2), primarily used for producing short-basalt fibres is a melt-blowing process. Here, molten basalt is poured on a top loading rotating cylinder (accelerating cylinder) and subsequently conveyed to the two underlying fibrillation shafts (fiberization cylinders) under tangential force. Due to high centrifugal forces, molten basalt detaches into small drops, then under the application of compressed air jets blown from nozzles (blowing valves) behind the fibrillation shafts, evolves into thin and cylindrical shapes, which after cooling leads to the formation of short basalt fibres [7,30,31].

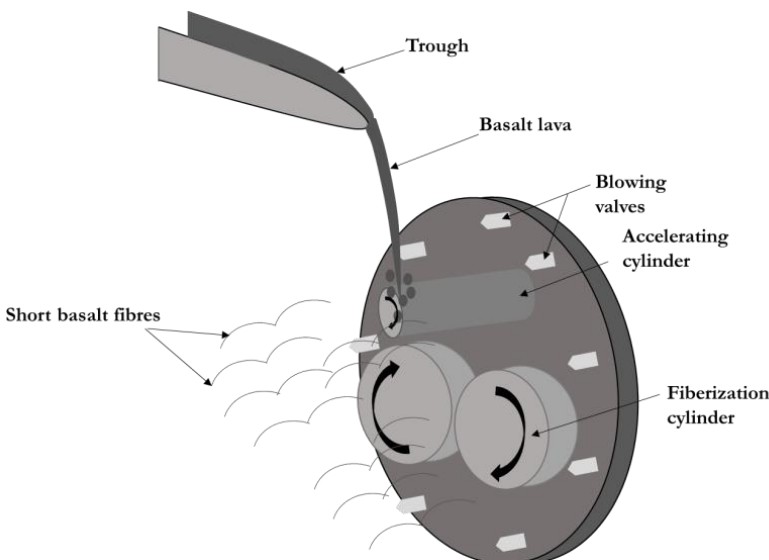

**Figure 2.** Schematic of Junkers technology (redrawn from [30,31]).

## 2.2. Mechanical Properties of Basalt Fibres

Basalt fibres generally have a linear elastic behaviour under tension until brittle failure [32,33]. Deák & Czigány [33] investigated the influence of geometrical features and chemical composition considering $SiO_2$ and $Al_2O_3$ content on tensile strength of basalt fibres from different manufacturers. The influence of an increase in $Al_2O_3$ content on the tensile strength of basalt fibre was investigated by Gutnikov et al. [34]. Both studies revealed that an increase in $Al_2O_3$ content from 10% to 24%, increased tensile strength of basalt fibres from 1.7 GPa to 2.5 GPa. The effect of the fibre manufacturing technology

and corresponding surface defects (such as microcracks, cavities) was investigated by Gurev et al. [35]. The presence of possible manufacturing defects does not affect fibre stiffness but has a significant impact on fibre tensile strength [35]. The properties of various fibres commonly used in composite materials for industrial applications are listed in Table 1. The cost of basalt fibre may rise due to supply difficulties after the current global political unrest but could fall well within the economical scale.

The effect of manufacturing technology, and surface defects on the mechanical properties of basalt fibres, optimizing by the fibre sizing (coupling agents) applied to basalt fibres during their production process was proposed by Greco et al. [32]. Sized basalt fibres demonstrated a significant improvement in mechanical properties over unsized fibres. A smooth surface was evident on basalt fibres coated with silane coupling agents [32]. Additionally, when embedded in a polymeric matrix the sized basalt fibres demonstrate improved adhesion characteristics than for sized glass fibres, being comparable to those of sized carbon fibres with the same polymeric matrix resin [32].

Typically, E-glass fibres are used as reinforcement in non-aerospace applications like wind turbine blades, for example, in the blade skins and shear webs of the main spar [36]. Carbon fibres are used in aerospace and automotive applications, and often used alongside E-glass fibres in the spar cap section of turbine rotor blades to increase the bending stiffness of the blade [37]. Alternatives to E-glass include glass fibres with modified compositions, such as S-glass (high strength glass). S-glass fibres exhibit approximately 40% higher tensile and flexural strength as well as 10–20% higher compressive and flexural modulus, compared to E-glass fibres [36,38]. However, S-glass is more expensive (~20%) than E-glass [36]. Carbon fibres have much higher stiffness and lower density than E-glass [38] but are more expensive [36,37,39,40]. Aramid fibres (sold commercially as Kevlar® and Twaron®) have higher tensile strength, impact properties, and superior damage tolerance to glass fibres. However, they have low compressive strength, absorb more moisture, and degrade under ultraviolet radiation [41]. A potential alternative to synthetic fibres like E-glass, can be basalt fibres [36]. Basalt fibres demonstrate better mechanical properties than most synthetic fibres like glass, aramid [9,36,42], though with a somewhat lower strain to failure (3.15% for basalt and 4.7% for E-glass [30]) and relatively high specific gravity (relative density) compared to E-glass (2.8 for basalt and 2.6 for E-glass) [9,30]. They are also significantly cheaper than carbon fibre [9,36,41,43].

**Table 1.** Comparison of properties of fibres commonly used in fibre reinforced polymer composites [9,30,38,43–51].

| Fibre | Fibre Diameter (μm) | Density (g/cm³) | Tensile Strength (MPa) | Modulus of Elasticity (GPa) | Elongation at Break (%) | Price (USD/kg) |
|---|---|---|---|---|---|---|
| Basalt | 9–23 | 2.8–3.0 | 3000–4840 | 79.3–93.1 | 3.1 | 2.5–3.5 |
| E-glass | 9–13 | 2.5–2.6 | 3100–3800 | 72.5–75.5 | 4.7 | 0.75–1.2 |
| S-glass | 9–13 | 2.46–2.5 | 4590–4830 | 88–91 | 5.6 | 5–7 |
| Carbon | 4–7.5 | 1.75–1.9 | 3500–6000 | 230–600 | 1.5–2.0 | 30 |
| Aramid | 5–18 | 1.44 | 2900–3400 | 70–112 | 2.8–3.6 | 25 |

### 2.3. Chemical Properties of Basalt Fibres

Basalt fibres exhibit good resistance to chemical degradation and moisture absorption, are non-combustible and possess good resistance to acidic or alkaline media [41,52]. In corrosive media basalt fibres retain better mechanical properties than glass fibres [53]. Indeed, the use of basalt fibres alongside carbon fibres as hybrid reinforcement for small wind turbine blades demonstrated encouraging results in terms of excellent fatigue strength, low weight, less cost and potential of recycling [36,54–56]. A comparison between the different chemical constituents of basalt and E-glass fibres is provided in Table 2.

An important difference between basalt and E-glass is the relatively large proportion of $Fe_2O_3$ (ferric oxide) in basalt [43,57], which, as it is a natural nucleating agent, imparts high thermal stability to basalt [57] and helps to maintain the crystalline structure. Epoxy resin is currently the most common matrix material used in conjunction with basalt fibres [43], but, with appropriate sizing, basalt is also compatible with resin systems such as phenolic, polyester and vinyl ester [7]. Ralph et al. [58] demonstrated that fibre sizing has a significant effect on the mechanical and chemical bonding of basalt fibres with a polymer matrix (especially for polypropylene). The chemical, mechanical and geometrical properties of basalt fibres were evaluated and compared with E-glass fibres by Ralph et al. [42]. The results revealed that basalt and E-glass fibre have similar elemental composition with basalt fibres demonstrating higher tensile strength than E-glass [42].

**Table 2.** Chemical composition of basalt and E-glass fibres [9,43,59].

| Oxides Content (wt. %) | Basalt | E-Glass |
|---|---|---|
| $SiO_2$ | 47.5–53.0 | 53.4 |
| $Al_2O_3$ | 13.3–18.0 | 14.3 |
| $Fe_2O_3$ | 7.0–14.0 | 0.28 |
| CaO | 8.0–11.0 | 19.0 |
| MgO | 3.5–5.0 | 3.3 |
| $B_2O_3$ | 0.8 | 10.3 |
| $TiO_2$ | 0.2–3.5 | 0.14 |
| $Na_2O + K_2O$ | 2.5–6.0 | 0.29 |
| $ZrO_2$ | 0.0 | 0.8 |
| MnO | 0.17–0.22 | N/A |

Chemical properties of basalt fibre when subjected to different alkaline or acidic components is important to understand the corresponding mass loss and strength reduction in mechanical properties of the fibre.

Acidic and alkaline resistance of basalt fibres was analyzed in several studies [13,60–62]. The studies revealed a better acid resistance (immersion in HCl solution) of basalt fibres when compared to exposure in an alkaline media (NaOH solution). Basalt fibres demonstrated ~8% mass loss (double that in acid media) in alkaline media and a degradation in tensile strength by ~35% as compared to 20% reduction in tensile strength in acid media [60]. The chemical stability of basalt fibres has been compared with that of glass fibres [13,61] and revealed an overall better acid resistance of basalt (10% fibre mass loss and 20% reduction in tensile strength) as compared to glass (30% reduction in fibre mass loss and tensile strength). Alkaline resistance of basalt fibres varied based on the exposure time [62]. With an increase in exposure time from 7 to 28 days, the fibre mass loss of basalt fibre increased from 20% to 70% and reduction in tensile strength changed from 50% to 80%, respectively [62]. In the same study, alkaline resistance of basalt fibres was compared with that of carbon fibres, which revealed a comparatively higher reduction in tensile strength (5% to 15%), and fibre mass loss (10% to 20%) for basalt fibres following immersion in an alkaline media over the same period as compared to carbon.

Degradation of basalt fibres following immersion in alkaline cement solutions has been investigated [63,64]. No significant reduction in fibre mass loss was reported for basalt fibres following immersion in cement solution [63,64]. Environmental degradation of basalt fibres following sunshine exposure was also studied by Sim et al. [62]. The tensile strength of basalt fibres reduced by ~13% following exposure to sunshine for 4000 h. which was similar to that observed for glass and carbon fibres [62].

### 2.4. Thermal Properties of Basalt Fibres

Basalt fibres demonstrate good thermal stability at elevated temperatures as compared to other fibres like glass, carbon [14,28,62,65,66]. Table 3 gives further insight into this information. Basalt fibres demonstrate high thermal stability over a wide range of temperature from −260 °C to 700 °C. This is mainly attributed to the material characteristics of basalt rocks that have a high nucleating temperature [62,66]. As a result of this, the

softening temperature of basalt fibres is at a temperature of around 960 °C, ~15% higher than E-glass [29].

**Table 3.** Thermal stability of different fibres [12,14,28].

| Fibre | Working Temperature Range ($\Delta T$) [°C] | Thermal Conductivity ($W \cdot m^{-1} \cdot K^{-1}$) | Thermal Expansion Co-Efficient ($10^{-6 \cdot} \, °C^{-1}$) |
| --- | --- | --- | --- |
| Basalt | −260 to 700 | 0.031–0.038 | 8.00 |
| E-glass | −50 to 380 | 0.034–0.040 | 5.40 |
| S-glass | −50 to 300 | 0.034–0.040 | 29.00 |
| Carbon | −50 to 700 | 5–185 (axial only) | 0.05 (axial only) |

The high thermal stability of basalt fibres is attributed to their crystallization behavior. Crystallization of a fibre primarily depends on the fibre chemical composition and heat treatment conditions. The crystallization of continuous basalt fibres during heat treatment was investigated by Moiseev et al. [67], and Gutnikov et al. [68] The high proportion of iron oxides initiates crystallization in basalt fibres with oxidation of ferrous cations and then formation of CaO, MgO, $(Mg,Fe)_3O_4$ nanocrystalline layers. Diffusion of divalent cations from the interior to the fibre surface leads to reaction with the environmental oxygen forming the nanocrystalline layers. Crystallization of basalt fibres can be controlled by doping with other elements for example with zirconium oxide [69]. ·

The literature [28,62,65] suggests good thermal stability of basalt fibres at temperatures around 200 °C following exposure for 1 hr demonstrating no significant effect on the tensile strength of fibres. Above 200 °C, the reduction in tensile strength of basalt fibres is slower than for other fibres like glass and carbon [28]. Basalt fibres also maintained good durability at temperatures between 600–1200 °C as compared to both glass and carbon [62]. Studies [28,65] suggest that thermal decomposition of basalt fibres initiates at a temperature of ~200 °C, and between 200–800 °C mass loss is ~0.74% as compared to glass fibres whose thermal decomposition starts at ~160 °C and between 160–850 °C demonstrates mass loss of about 1.8% [65].

## 3. Mechanical Properties of Basalt Fibre Reinforced Composites

Continuous basalt fibres in different fabric architectures such as woven or non-woven can be embedded in a polymeric resin (for example, epoxy, polyester, vinyl ester thermosets or thermoplastics) to form FRC materials. These FRC materials can consist of one or more plies arranged in specific stacking sequences to form FRC laminates.

Several studies [70–80] in the literature have focused on improving the fibre/matrix adhesion properties to improve the mechanical response of basalt FRC. Different fibre surface treatment strategies were adopted to improve the fibre/matrix interfacial bond strength. Strategies included colloidal silica and sol/gel techniques [70,71], silanized and acid treated multi-walled carbon nanotubes [72–76], increasing mechanical interlocking by incorporating silica nanoparticles [77], plasma treatment of basalt fibres to increase fibre surface roughness and adhesion characteristics [78,79], and resin hybridization [80] were adopted in the respective studies. These techniques significantly improved mechanical properties of basalt FRC. For example, Wei et al. [71] reported that the application of SiO2 nanoparticles by a sol-gel technique improved tensile strength of basalt FRC by ~30% and ILSS increased by ~15% [71]. Kim et al. [73] reported that the application of silanized and acid-treated carbon nanotubes both enhanced the flexural properties and fracture toughness of basalt/epoxy composites. SEM examination demonstrated improved flexural properties and fracture toughness in silanized carbon nanotube treated basalt/epoxy composites as compared to the basalt/epoxy composites with acid-treated carbon nanotubes [73]. Application of acid treated carbon nanotubes significantly enhanced the wear properties of basalt/epoxy composites [74]. Wei et al. [77] reported that silica nanoparticle-epoxy coating significantly improved tensile strength of basalt fibres as compared to pure epoxy coating. The silica nanoparticle-epoxy coated basalt FRC also demonstrated significant

improved interfacial properties [77]. Kim et al. [79] reported that low-temperature oxygen plasma treatment of basalt/epoxy composites demonstrated a significant improvement in interlaminar fracture toughness. SEM micrographs revealed a good adherence between resin and fibres for plasma treated basalt/epoxy specimens as compared to untreated ones [79]. Dorigato & Pegoretti [80] investigated resin hybridization and treatment of fibre surface for the mechanical and failure behaviour of basalt fibre mat-reinforced composites. Two coupling agents constituting epoxy and vinylester functional groups were embedded to improve the interfacial characteristics of basalt fibres with the resin system. Resin hybridization was achieved by incorporating vinylester/epoxy (VE/EP) ratios at 1:1, 1:3 and 3:1, respectively. The study demonstrated a significant improvement in mechanical strength and fracture toughness of basalt/epoxy composites manufactured using a hybrid resin system constituting VE/EP matrix at a ratio of 1:1 reinforced with surface treated basalt fibres [80].

The mechanical properties of basalt FRC materials are now summarized and discussed [80–86] Basalt FRC have comparatively higher (or comparable) mechanical properties when compared to glass FRC materials with similar fibre volume fraction and using the same polymeric resin. In each of the studies [80–84,86], epoxy was the primary resin matrix, while vinyl ester resin was used by Carmisciano et al. [85].

Dorigato & Pegoretti [80] evaluated the tensile properties of plain woven (PW) basalt FRC materials and compared with a glass-based version. With ~60% fibre volume fraction, basalt FRCs demonstrated a 17% and 20% higher tensile strength and elastic modulus, respectively, than glass FRC with similar fibre volume fraction. In contrast, the tensile strength of basalt FRC was ~14% lower than their carbon counterparts with similar fibre volume fraction [80]. Unidirectional (UD) basalt FRC materials demonstrated comparatively higher failure strain as compared to both UD glass (~5% lower) and carbon FRC (~30% lower) materials [81]. PW basalt FRC demonstrated ~40% higher compressive strength and modulus as compared to PW glass based FRCs [82]. Palmieri et al. [83] reported that basalt FRC found ~20%, 30% and 12% higher tensile strength, elastic tensile modulus and failure strain as compared to glass FRC. Tensile and compressive strengths of PW fabric reinforced basalt FRC were ~23% and 43% higher than PW glass fabric reinforced FRC [84]. Carmisciano et al. [85] reported that flexural strength and modulus of PW fabric reinforced basalt FRC were ~20% higher than glass-based ones.

The influence of matrix properties on mechanical properties of basalt FRC was evaluated by Černý et al. [86]. Basalt FRC manufactured using epoxy resin matrix demonstrated higher tensile strength and modulus (20% and 15% higher, respectively) as well as higher compressive strength by ~45% as compared to those manufactured using vinyl ester resin matrix.

Interlaminar shear strength (ILSS), flexural strength and modulus of PW fabric reinforced basalt FRC under 3-point bend loading has been analysed by multiple researchers [82,85–93]. A comparison of flexural strength, and ILSS of basalt FRC against composites manufactured using other types of reinforcement, such as, E-glass is illustrated in Figures 3 and 4.

The reported flexural strength, flexural modulus and interlaminar shear strength varied not only based on the material system but also strongly on the manufacturing method used which include hand lamination, vacuum assisted resin infusion (VARI), and resin transfer moulding. PW basalt/epoxy composites were manufactured using vacuum assisted resin transfer moulding (VaRTM) in studies [87,89,90,92]. VaRTM is also referred to as Resin infusion under flexible tooling with a flow medium (RIFT -II) [38] outside North America. In other studies, hand lamination using an impregnation roller technique followed by hot mould press [88] or resin transfer-moulding using a fixed cavity mould [91] was used for manufacturing PW basalt/epoxy composites.

Flexural strength values of basalt FRC varied from ~229 MPa as reported by Subagia et al. [87] to 505 MPa reported by Lopresto et al. [82]. Flexural modulus varied from ~4.8 GPa as reported by Bulut [88] to 23 GPa reported by Lopresto et al. [82]. ILSS was reported in [82,84,92,93]. ILSS values varied from ~18.9 MPa for PW basalt FRC manu-

factured by VARI using hand roller technique as reported by Lopresto et al. [82] through 41 MPa for basalt FRC manufactured using VaRTM technique reported by Scalici et al. [92] to 60 MPa for bi-directional woven basalt/epoxy composite reported by Dorigato & Pegoretti [93]. Laminates were manufactured using positive pressure infusion of 1 MPa in a fixed cavity mould and cured under pressure and vacuum for 2 h at 50 °C and 2 h at 80 °C [93]. PW basalt/epoxy composites have been compared to PW E-glass using VaRTM. The flexural strength and ILSS were found to be up to 55%, and 50% higher, respectively, for the basalt/epoxy composite as reported by Lopresto et al. [82], and Chairman et al. [84]. In contrast, the ILSS of PW basalt fabric/ epoxy-based vinyl ester resin was ~20% higher than an E-glass based laminate, but flexural strength was lower by ~15% as reported by Carmisciano et al. [85].

Impact damage mode and mechanical properties of UD basalt fibre reinforced epoxy composites was investigated by He et al. [94]. Comparisons were drawn against composites manufactured using S-2 glass and aramid. Hot-press moulding technique was used to manufacture composite laminates and a fibre volume fraction of ~60% was achieved for each type of composites manufactured. Impact damage was characterized by Charpy-impact testing procedure and post impact properties were analysed by performing a 3-point bend test. For basalt and aramid fibre reinforced composites, damage evolution was progressive evolving layer-by-layer, whereas glass-based composites demonstrated brittle failure. The reduction in flexural properties between the back face and the impact face of the composites was highest in aramid-based composites followed by glass and basalt demonstrating the least variation [94]. Sfarra et al. [95] demonstrated a comparison of damage features by impact testing for basalt and glass fibre reinforced epoxy composites. The authors found more anisotropic impact damage behaviour in basalt composites compared to glass fibre composites, which can produce a limitation on predicting the mechanical behaviour of basalt fibre composites [95]. Shishevan et al. [96] investigated the low velocity impact behavior of twill-woven basalt fibre reinforced epoxy composites and compared the impact key parameters with Carbon fibre reinforced composites. VaRTM technique was used to fabricate composite laminates and fibre volume fraction of ~ 60% was achieved for each type of composites. Low velocity impact response of basalt/epoxy composites were investigated by related force-deflection, force-time, deflection-time, and absorbed energy-time graphs. Basalt/epoxy composites demonstrated better low velocity impact performance than carbon/epoxy composites [96]. Impact properties of plain-woven and unidirectional basalt epoxy composites were investigated by Fu et al. [97]. Low velocity impact and ballistic tests were performed to characterize the impact properties of both type of composites. Under low velocity impact test with a hemispherical impactor, unidirectional basalt/epoxy composites demonstrated a higher impact resistance as compared to woven basalt/epoxy composites, but under a sharp impactor, the impact response of unidirectional basalt/epoxy composites were lower than woven basalt/epoxy composites. In ballistic tests, the ballistic property of unidirectional basalt/epoxy composites were higher than that of woven basalt/epoxy ones [97]. Sanchez-Galvez et al. [98] investigated the performance of neat basalt, hybrid glass/basalt, and hybrid carbon/basalt vinylester composites under high-speed impact tests and by comparing their ballistic limits. The best performance was observed for hybrid glass/basalt vinylester composite which demonstrated the highest ballistic limit (~ 480 m/s) [98]. Low-velocity impact performance of basalt/polyester composites accounting to the variation in the number of basalt fibre layers through-thickness was investigated by Arunprasath et al. [99]. Samples with smaller number of basalt fibre layers (1–4) failed rapidly demonstrating extensive regions of delamination. However, samples with relatively greater number of basalt fibres (5–9) demonstrated higher impact resistance and a progressive damage behavior. In another study, Dhakal et al. [100] investigated falling weight impact damage characteristics of plain-flax and falx/basalt hybrid vinylester composite. The experimental results demonstrated superior high impact energy and peak load of hybrid flax/basalt composites as compared to plain flax/vinylester ones demonstrating the hybridization technique as a promising strategy to improve toughness

properties of natural fibre reinforced composites [100]. Zuccarello et al. [101] studied the influence on mechanical performance and ageing characteristics of sisal-reinforced bio composites following hybridization with basalt fibres. The experimental analysis revealed improvement in mechanical performance and significant reduction in ageing effects on the mechanical properties of hybrid sisal/basalt fibre-reinforced bio composites with increase in basalt fibre volume fraction in the bio composite [101]. Mechanical performance of hybrid bast/basalt fibre reinforced polymer composites was investigated by Saleem et al. [102], where addition of basalt fibres improved mechanical properties and energy absorption capacity of the composite which is a key requirement in the automotive sector.

Bending behaviour of timber beams with composite reinforcements was analysed by De La Rosa García et al. [103]. Timber beams with UD basalt composites at surfaces demonstrated comparatively higher strength and stiffness than timber beams with carbon fibre-based reinforcements [103]. Mechanical properties of basalt FRC using non-crimp fabric (NCF) reinforcement have been reported in the literature [104,105]. Laminates were manufactured using VaRTM technique. ILSS of NCF basalt/epoxy composites was reported to be 44 MPa whereas the flexural strength and modulus were reported to be 698 MPa and 38.4 GPa, respectively, as demonstrated by Davies et al. [104]. NCF basalt/epoxy composites outperformed NCF E-glass/epoxy by ~15% in terms of flexural strength but ILSS of NCF E-glass/epoxy was ~8% higher than that of NCF basalt/epoxy thus demonstrating a relatively poor interfacial strength [104]. Chowdhury et al. [105] tested the flexural strength of NCF basalt epoxy composites in situ under a scanning electron microscope and compared with NCF E-glass/epoxy composites with similar fibre volume fraction (~54%). NCF basalt/epoxy composites demonstrated ~10% higher flexural strength as compared to NCF E-glass/epoxy composites [105].

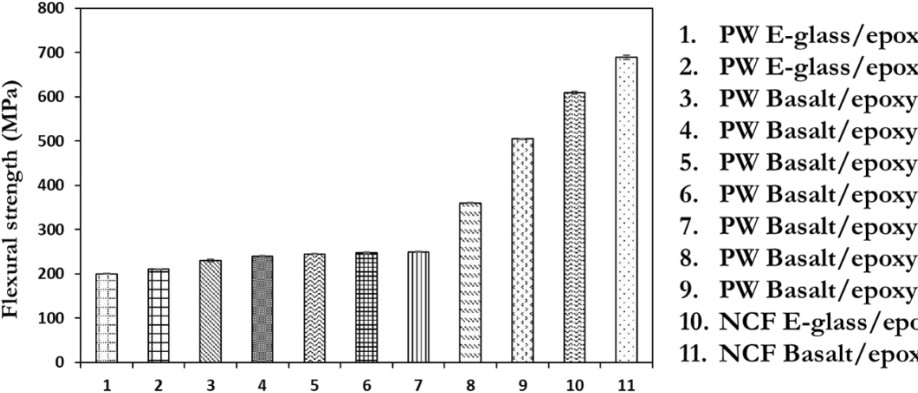

**Figure 3.** A comparison of flexural strength of Basalt/epoxy composite against composites manufactured using E-glass based reinforcement [1—Scalici et al. [92], 2—Lopresto et al. [82], 3—Bulut [88], 4—Sarasini et al. [91], 5—Ary Subagia et al. [87], 6—Petrucci et al. [89], 7—Scalici et al. [92], 8—Sun et al. [90], 9—Lopresto et al. [82], 10, 11—Fu et al. [97]] [PW < NCF (0/90)].

Alongside epoxy, vinylester and polyester resin have also been used as the polymeric resin matrix to embed basalt fibre composites. De Rosa et al. [106] compared post-impact performance of PW basalt and E-glass FRC. Basalt- and glass-fibre composites demonstrated similar damage tolerance to impact, but post-impact residual properties were superior for basalt composites [106]. Impact behaviour of basalt fibre reinforced unsaturated polyester resin composites was investigated by Gideon et al. [107] by performing static 3-point bending and low velocity impact tests. PW, cross-ply and UD basalt composites were manufactured by hand lay-up and hot pressed under pressure techniques. UD basalt composites demonstrated superior mechanical properties under static loading, while woven and cross-ply laminates outperformed UD basalt composites under dynamic loading conditions. A direct correlation between the effect of fabric architecture, fibre lay-up and testing parameters on the mechanical performance of basalt composites was revealed [107].

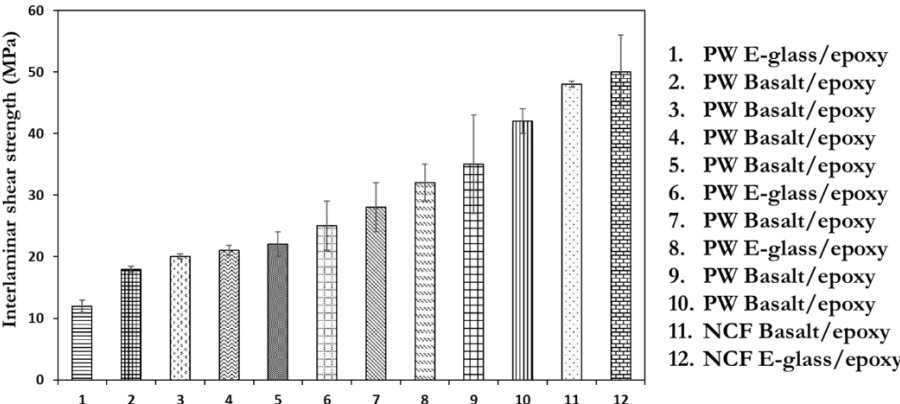

**Figure 4.** A comparison of interlaminar shear strength of Basalt/epoxy composite against composites manufactured using E-glass based reinforcement [1—Chairman et al. [84], 2—Carmisciano et al. [85], 3—Lopresto et al. [82], 4—Sarasini et al. [91], 5—Ary Subagia et al. [87], 6—Chairman et al. [84], 7—Bulut [88], 8—Lopresto et al. [82], 9—Sun et al. [90], 10—Petrucci et al. [89], 11, 12—Fu et al. [97]] [PW < NCF (0/90)].

The literature has also focused on investigating mechanical properties of thermoplastic polymer matrix composites with basalt fibres as the reinforcement. Basalt fibre reinforcement improves the structural integrity, quasi-static mechanical properties [108], friction and wear behaviour [109–111], dynamic mechanical properties and injection moulding shrinkage [112], tensile, and flexural properties as well as dispersion of nanoparticles [113] in thermoplastic matrix composites.

*Failure Mechanisms in Basalt Fibre Reinforced Composites*

Common types of fabric architectures widely used to manufacture basalt FRC materials are UD, bidirectional, PW, non-crimp fabric (NCF), short discontinuous fibre reinforcement. Figure 5 shows schematic representations of commonly used fabric reinforcements for manufacturing basalt fibre reinforced composites is illustrated.

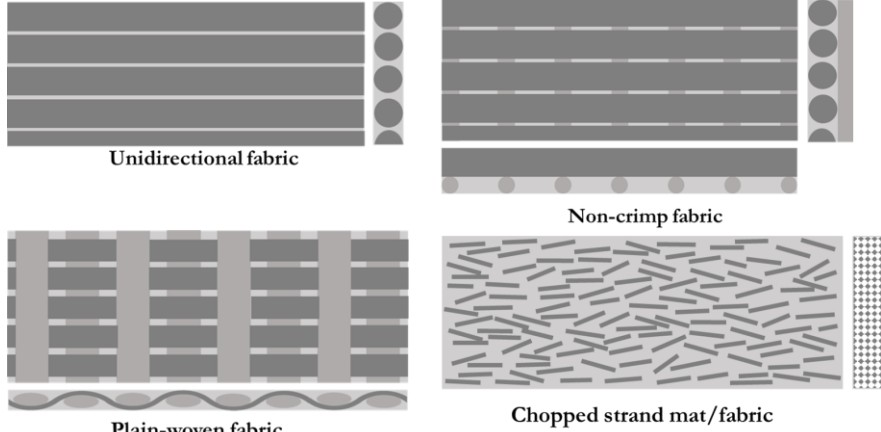

**Figure 5.** Commonly fabric architectures used to manufacture basalt fibre reinforced composites.

FRC generally exhibit a combination of different failure mechanisms depending on the loading conditions [114–116]. Different factors contributing to the failure behaviour in FRC include the anisotropic nature of each ply, ply orientation, fibre architecture, fibre/matrix interfacial bond strength, interaction between different failure modes and loading direction. Each of these factors contribute at each ply level leading to the overall composite structure having a range of failure modes under different loading conditions [117].

Typical damage mechanisms observed in UD basalt FRC with fibres aligned parallel to the tensile axis of the composite structure include: (i) fibre splitting/breakage along

the tensile axis under tensile loading [118,119], (ii) fibre buckling under compressive loading [120], and/or (iii) failure at the fibre/matrix interface under shear loading, leading to matrix cracking in a direction generally parallel to the fibre tensile axis [121].

When fibres are aligned transverse to the loading direction different failure modes are observed: (i) failure at the fibre/matrix interface under tensile loading, continuing through the thickness with increasing load leading to fracture in the direction perpendicular to the loading direction [122], (ii) shear failure at an angle of ~45° to the loading direction under compressive loading [122], (iii) matrix yielding in the plane perpendicular to the fibres under shear loading, leading to crack initiation at the fibre/matrix interface and eventually formation of a crack within the ply at an angle of ~45° to the tensile axis [117,122].

Damage in NCF basalt FRC has been observed to initiate within the 90° sub-plies or at the point of contact between the 0° UD and 90° sub-plies or at the resin rich volumes (RRV) [123] present between two corresponding NCF based plies [124,125]. The 0° and 90° sub-plies refer to the corresponding sub-sections of a NCF ply. Typical failure modes observed on the compression side in a NCF basalt fibre reinforced epoxy composite under bending loads are fibre kinking in 0° sub-ply, intra-ply delamination, and/or transverse matrix crack across the 90° sub-ply. The corresponding failure modes observed on the tension side of a NCF basalt fibre reinforced epoxy composite under similar loading conditions is fibre breakage in the 0° sub-ply, intra-ply delamination, and/or transverse matrix crack across 90° sub-ply. Typical failure modes observed in NCF basalt/epoxy composite under bending loads are further illustrated in Figures 6 and 7.

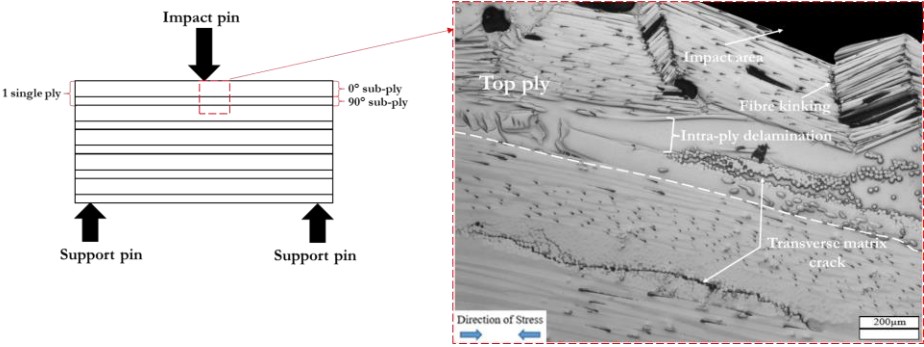

**Figure 6.** Typical failure modes on the compression side of a NCF basalt/epoxy composite under bending loads.

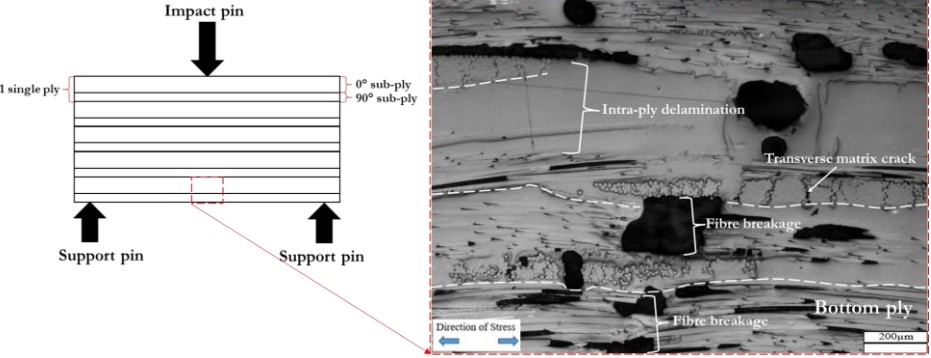

**Figure 7.** Typical failure modes on the tension side of a NCF basalt/epoxy composite under bending loads.

The most commonly observed failure modes for PW basalt FRC under bending loads are fibre pull-out on the tension side, fibre-matrix debonding [87,88], fibre buckling and kink bands on the compression side [88,91] and fibre breakage (fibre fracture and pull-out) on the tension side [82,90].

## 4. Durability of Basalt Fibre Reinforced Composites

Durability of a material is determined by its resistance to damaging effects such as extreme temperature, moisture, ultra-violet radiation, exposure to aggressive chemicals, stress cycle (fatigue). Durability is assessed by measuring material property such as strength, modulus, before and after exposure to one or more damaging effects over a period under predefined conditions.

### 4.1. Durability in Moist Conditions

Structural composites are often exposed to environmental moisture during their service life period. Generally, moisture absorption in FRC takes place through three different mechanisms (see Figure 8): (i) diffusion of moisture content through micro-gaps (free volume) between polymer chains, (ii) capillary transport of water into the micro-gaps, and (iii) through flaws at the interface between fibre and matrix [126–128]. Moisture ageing imparts morphological changes in fibre reinforced polymer composites in various ways. Firstly, matrix swelling can induce internal stresses leading to micro-crack formation, then diffusion paths thus altering the moisture absorption and diffusion characteristics of the composite. Secondly, plasticization of the matrix can lead to an increase in strain-to-failure of the composite and reduction in glass transition temperature (Tg) (Smith & Schmitz [129] proposes that moisture absorption of ~2% can lead to a 20% reduction in glass transition temperature of a typical polyester resin). Thirdly, the chemical bonding at the fibre/matrix interface can be affected by moisture which in turn impacts the strength and stiffness of the composite [130]. Wright [128] plotted the fall in glass transition temperature (Tg) as a function of moisture content for data from epoxy resins (from five separate published papers) and found "as a rough rule-of-thumb" that there was a drop of 20 °C for each 1% of water pick-up (data available up to 7% moisture content). For saturated PMMA at 1.92% water pick-up, the Tg was depressed by ~20 °C [129]. For PLA microspheres, Tg was reduced from 52 °C (~0.3% $H_2O$) to 37 °C (3.5% $H_2O$), implying a need for the cautious design of PLA matrix composites to be used in humid tropical environments [130].

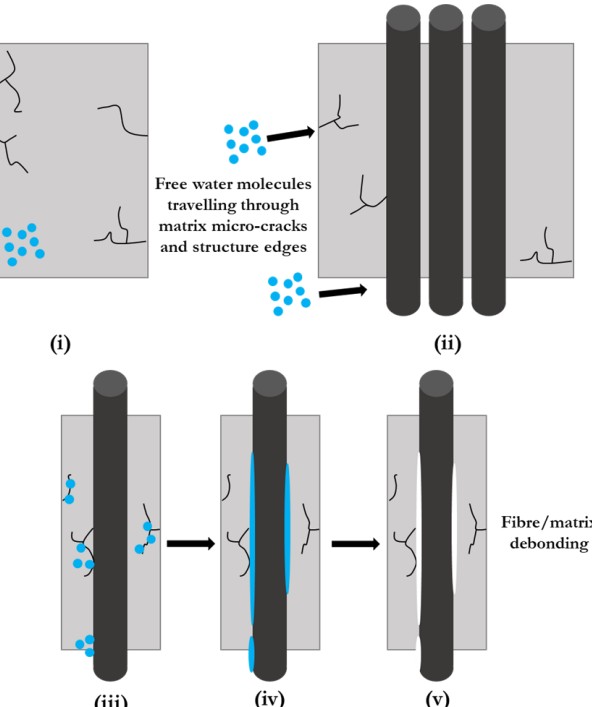

**Figure 8.** Mechanism of moisture absorption: (**i,ii**) free water molecules travelling through matrix and micro-cracks, (**iii–v**) moisture diffusion through matrix and microcracks (redrawn from [127]).

Swelling of materials is measured by a Coefficient of Hygroscopic Expansion (CHE, β), also known as the swelling- or moisture- expansion coefficients and defined as the strain, $\varepsilon$, induced by a variation of 1% of moisture content (ΔM). Anisotropic materials will have different CHE in each direction, but for natural fibres there will be just a longitudinal (axial) and transverse (radial) CHE.

The moisture absorption of a material can be quantified by the diffusion coefficient, $D$, and moisture absorption capacity, $M^\infty$. Typically, $D$ and $M^\infty$ are evaluated at temperatures at least 20 °C below the glass transition temperature of the material under test [131,132]. $D$, which quantifies the rate of moisture uptake and is generally based on Fickian diffusion theory [133], is temperature dependent, generally increasing with increasing temperature [134].

Davies et al. [104] found $D$ (see Figure 9) for NCF basalt/epoxy increased from $2.4 \times 10^{-12}$ m$^2$/s at 4 °C to $79 \times 10^{-12}$ m$^2$/s at 60 °C; for NCF E-glass/epoxy, $D$ increased from $1.8 \times 10^{-12}$ m$^2$/s at 4 °C to $96 \times 10^{-12}$ m$^2$/s at 60 °C. It is evident that NCF E-glass/epoxy demonstrates relatively higher $D$ (~20% higher) compared to NCF basalt/epoxy with increasing temperature. A detailed comparison of the variation in diffusion coefficient, $D$, of NCF Basalt and E-glass data from Davies et al. [104] is presented in Table 4 where a dependency of $D$ on temperature, and material type is further evident. Both composites have a vacuum infused Araldite LY 1564 epoxy with Aradur 3687 hardener matrix.

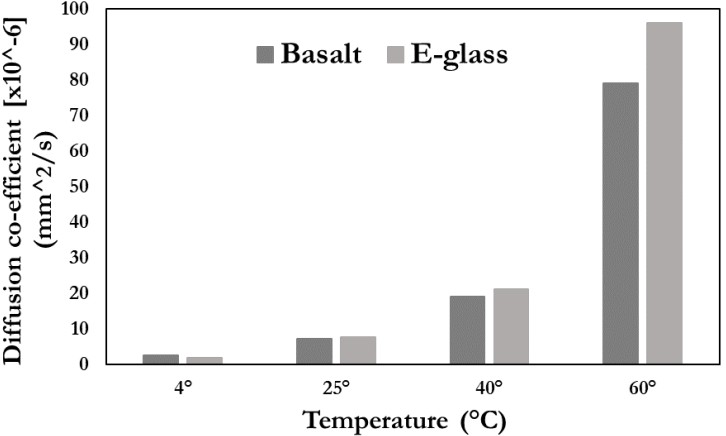

**Figure 9.** Comparison of D between NCF basalt/epoxy and NCF E-glass/epoxy at different temperatures (plotted from [104]).

**Table 4.** An overview of the variation in diffusion coefficient, D of NCF Basalt and E-glass data obtained from the literature [104] following immersion in seawater.

| Material | | $\mathbf{D} \times 10^{-12}$ **(m$^2$/s)** | | | |
|---|---|---|---|---|---|
| Fibre type | Lay-up | 4 °C | 25 °C | 40 °C | 60 °C |
| Basalt | 0° UD | 2.4 | 7 | 19 | 79 |
| E-glass | Quasi-UD | 1.8 | 7.5 | 21 | 96 |

$M^\infty$ for a composite is the maximum volume of moisture, expressed as weight percent, that the material can absorb and is dependent on the specimen dimensions, matrix type, fibre layup and the manufacturing process [134]. An overview of $M^\infty$ of basalt FRC materials from the literature is presented in Table 5 where a strong dependency of $M^\infty$ on the manufacturing process, resin, fibre lay-up and conditioning environment (media and temperature) is evident.

**Table 5.** An overview of the variation in moisture absorption capacity, $M^{\infty}$. data obtained from the literature when different types of reinforcement, epoxy resin, manufacturing technology and conditioning environment has been used (PW—Plain-woven, NCF—Non-crimp fabric, UD—Unidirectional).

| Material | NCF Basalt | NCF E-Glass | PW Basalt | PW Basalt | PW E-Glass | UD Basalt | UD Basalt | UD E-Glass |
|---|---|---|---|---|---|---|---|---|
| Epoxy | Araldite 1564 LY | Araldite 1564 LY | Polyester | Epoxy RIM 135/137 | Epoxy RIM 135/137 | Epoxy JN-C3P | Epoxy Bisphenol-A | Epoxy Bisphenol-A |
| Manufacturing process | VaRTM | VaRTM | Compression moulding | VaRTM | VaRTM | VaRTM | Pultrusion | Pultrusion |
| Immersion period | 200 days | 200 days | 24 h | ~100 days | ~100 days | 45 days | 84 days | 84 days |
| Temperature | 40 °C | 40 °C | Ambient | 80 °C | 80 °C | 40 °C | 40 °C | 40 °C |
| Media | seawater | seawater | seawater | distilled water | distilled water | distilled water | seawater | seawater |
| $M^{\infty}$ | 1.5% | 1.25% | ~2% | ~3.5% | ~6% | ~0.8 % | ~3% | ~0.3% |
| Literature | [104] | [104] | [135] | [136] | [136] | [137] | [138] | [138] |

Numerous moisture absorption studies evaluating the weight gain characteristics ($M^{\infty}$) of basalt/epoxy composites have been performed [104,135–138]. NCF based basalt/epoxy composites, manufactured using VaRTM and immersed in seawater for 200 days at 40 °C, demonstrated a weight gain of 1.5%, while for E-glass/epoxy composites under the same conditions the weight gain was 1.25% [104]. PW basalt/polyester composites manufactured using compression moulding aged in normal water and seawater for 24 h demonstrated a similar weight gain of ~2% in both media [135]. PW basalt/epoxy composites manufactured using VaRTM, demonstrated a weight gain of ~3.5% following an immersion period of ~100 days at 80° C in distilled water. The weight gain for PW E-glass/epoxy composites under the same conditions was ~6% [136]. UD basalt/epoxy composites, manufactured using VaRTM, demonstrated a weight gain of ~0.8% following 45 days immersion in distilled water at 40 °C [137]. Pultruded UD basalt/epoxy demonstrated a weight gain of ~3% after ageing in seawater at 40 °C for a period of 84 days, compared to pultruded UD E-glass/epoxy which demonstrated a weight gain of ~0.3% under same conditions [138].

From these observations, it can be seen that different epoxy matrices, fibre architecture, manufacturing technique and conditioning environments utilised in each of the studies may have influenced the moisture absorption capacity of the composites. When composites are exposed to hygrothermal ageing conditions, moisture diffuses through the matrix. This results in matrix swelling and an increase in fluid pressure locally, which can lead to distortion or delamination [139,140].

Davies et al. [104] investigated the flexural strength and interlaminar shear strength (ILSS) of fully saturated NCF basalt/epoxy and NCF E-glass/epoxy composite laminates, manufactured using VaRTM, following seawater immersion for 200 days at 25 °C. NCF basalt/epoxy composites demonstrated a 25% reduction in flexural strength following seawater immersion compared to NCF E-glass/epoxy composites, which demonstrated a 6% reduction [104]. The ILSS of NCF basalt/epoxy decreased by ~22% compared to ~20% reduction for NCF E-glass/epoxy [104]. Failure modes observed for the flexural samples using SEM were compression and delamination on the compression side while there was no evidence of tensile failure modes [104]. However, this study only presented the failure modes observed at failure and did not focus on presenting a detailed analysis of initiation and propagation of failure modes in NCF basalt/epoxy and NCF E- glass/ epoxy composites.

Pandian et al. [135] reported that flexural strength of PW basalt/epoxy composites was reduced by ~42% and 48%, respectively, following immersion in normal water and seawater for 24 h [135]. The effect of moisture absorption on the ILSS of PW basalt/epoxy and E-glass/epoxy composites following immersion in distilled water at 80 °C for 100 days was evaluated by Kim et al. [136]. The ILSS of basalt/epoxy composites was reduced by ~40% compared to ~60% reduction observed in E-glass/epoxy [136]. The reduction in ILSS was attributed to the weakening of the fibre/matrix interfacial strength following moisture

ageing [129]. The ILSS of basalt twill fabric/epoxy composites, manufactured using hand lamination with an impregnated roller, demonstrated a drop of ~40% following ageing in salt water for 240 days [141].

However, E-glass twill fabric/epoxy composites demonstrated no significant drop in ILSS under the same conditions [141]. The effect of seawater on PW basalt/epoxy and E-glass/epoxy was investigated by Wei et al. [142]. The flexural strength of basalt/epoxy laminates, manufactured using the hot-press moulding process, decreased by ~40% following immersion in artificial seawater for a period of 90 days at 25 °C. The percentage degradation in flexural strength for E-glass/epoxy was lower (~30%) under the same conditions [142]. A comparison between flexural strength and interlaminar shear strength of basalt fibre reinforced composites against E-glass fibre reinforced composites following moisture-ageing is shown in Figures 10 and 11. No distinct change in flexural strength and modulus was observed for UD prepreg basalt/epoxy composites, manufactured utilizing hot-press moulding process, following immersion in distilled water for 90 days at room temperature [60]. The resistance to moisture ageing was attributed to the fibre/matrix interface not being significantly affected by distilled water [60]. The literature reports that moisture ageing negatively affects flexural and ILSS of fibre reinforced polymer composites, and the severity of degradation varies depending on the type of reinforcement, manufacturing technique and conditioning environments.

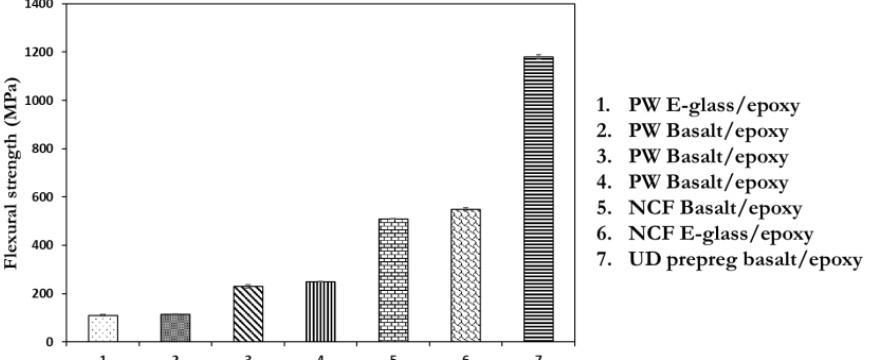

**Figure 10.** A comparison between flexural strength following moisture ageing of Basalt/epoxy composites against composites manufactured using E-glass fibre as reinforcement [1,2—Wei et al. [142], 3,4—Pandian et al. [135], 5—Chowdhury et al. [12], 6—Davies et al. [104], 7—Wang et al. [138]] [PW < NCF (0/90) < UD].

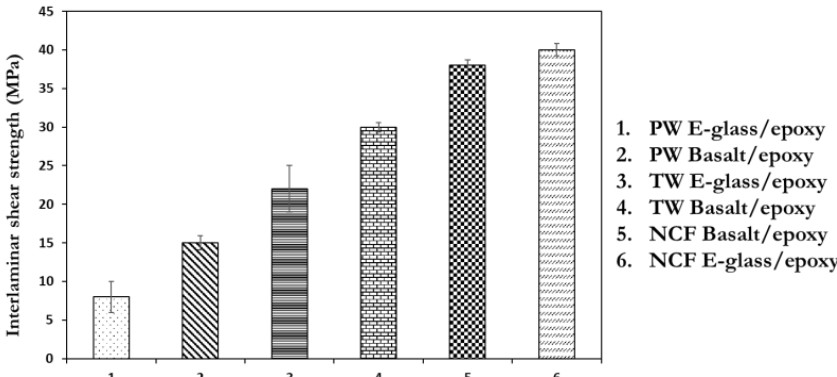

**Figure 11.** A comparison between interlaminar shear strength following moisture ageing of Basalt/epoxy composites against composites manufactured using E-glass fibre as reinforcement [1,2—Kim et al. [136], 3,4—Liu et al. [141], 5—Chowdhury et al. [12], 6—Davies et al. [104]] [PW < TW < NCF(0/90)].

### 4.2. Thermal Stability

For industrial applications, where threat of fire is inevitable, thermal stability of industrial composites is crucial to reduce casualties and property damage. Thermal stability of a material refers to all physico-chemical changes during combustion of the material which is generally measured by limiting oxygen index (LOI), heat release, smoke, mass loss, carbonization indicators. Tang et al. [143] reported that basalt fibre reinforced polypropylene composites demonstrated lower LOI (reported at 18.6) than pure polypropylene (reported at 19.1). At the same oxygen concentration, basalt fibre reinforced polypropylene composite took more time to burn as compared to pure polypropylene and demonstrated a better anti-melt dropping effect. Thermogravimetric analysis demonstrated that basalt fibre could reduce the maximum thermal decomposition rate and increase the temperature of maximum thermal decomposition rate of polypropylene. Further from the results of cone calorimeter test (CCT) it was revealed that heat release rate (HRR), total heat release (THR), rate of smoke release (RSR), and total smoke release (TSR) of basalt fibre reinforced polypropylene composite were lower than polypropylene. Overall, basalt fibre reinforced polypropylene composite demonstrated higher thermal stability and combustion performance than pure polypropylene [143].

Thermal stability of FRC manufactured using continuous basalt fibres following partial pyrolysis of polysiloxane matrix was investigated by Cerný et al. [86]. Composites were manufactured by wet-winding method, and then further pyrolyzed in nitrogen atmosphere between 650–750 °C. It was demonstrated that the thermal effects were more dependent on matrix governed elastic properties, such as shear modulus, as compared to Youngs modulus of the composite which is dominated by reinforcing fibres [86]. Landucci et al. [144] investigated the suitability of basalt based FRC as passive fire protection for jet fires at laboratory scale. Results demonstrated a comparatively higher softening temperature of basalt than glass, in addition to reduced mass loss indicating better suitability of basalt fibre-based composites for fire protection devices [144]. Thermal stability of basalt based FRC under the effect of freeze-thaw cycles have been investigated [145,146]. Basalt based FRC demonstrated a superior thermal stability characterized by higher retention rates of mechanical properties than both glass and carbon based FRC.

### 4.3. Chemical Durability

Service life of composites can be reduced due to exposure to corrosive agents. Exposure of composites to corrosive environment can induce significant degradation of its material mechanical behaviour. Chemical durability of basalt based FRC in different kinds of chemical solutions was investigated by Mingchao et al. [60]. Chemical solutions constituting 30% vitriol, 5% hydrochloric acid, 5% nitric acid, 10% sodium hydroxide, saturated sodium carbonate solution, 10% ammonia, acetone and distilled water was used to investigate the chemical durability of basalt based FRC, which demonstrated better chemical durability and corrosive behaviour in alkaline solutions [60].

Stress-corrosion behaviour of basalt based FRC was investigated following immersion in 5% sulphuric acid based corrosive medium [147]. Interrupted tests were performed at 30%, 50%, and 70% of ultimate strength. Results demonstrated a time dependent degradation behaviour of basalt FRC materials which accelerated at stress states more than 50% of ultimate strength. Following SEM observation, extreme effects of plasticization of the matrix on the composite specimens were observed [147].

## 5. Life Cycle Assessments (LCA)

LCA are one of the most common techniques which can be used to evaluate the environmental impact of FRC materials in consideration. FRC materials used in different industrial applications are generally manufactured using conventional fibre reinforcements, such as carbon, E-glass, which has anthropogenic environmental impact. The growing awareness about global warming and climate changes motivated researchers to evaluate the environmental impact associated with production of FRC materials. LCA analysis

enables the researchers to evaluate the environmental impact of FRC materials from the stage of retrieving the raw material until the products are recycled or wasted (Cradle-to-Grave [148]).

Table 6 presents LCA analysis of environmental externalities associated with the production of 1 tonne of continuous basalt or of glass fibres. Here, a Cradle-to-Gate LCA approach [148] is undertaken to evaluate the environmental impact associated over the course of production of 1 tonne of basalt fibre which has been compared with same functional unit for Glass fibre production. Among different categories, it is clearly evident that basalt fibre production has relatively much lower impact in terms of ozone layer depletion (~90% lower), and global warming (~80% lower) than glass fibre production. A major issue with LCA is freedom to use different functional units, goal, and scope/system boundaries and to report data in incomparable formats.

**Table 6.** Environmental externalities associated with production of 1 ton of basalt and glass fibre [149,150].

| Category | Abbreviation | Unit | Basalt Fibre [149] | Glass Fibres [150] |
|---|---|---|---|---|
| Source | | | | |
| Carcinogens | AC | kg $C_2H_3Cl$ eq | 15.2 | - |
| Non-Carcinogens | NC | kg $C_2H_3Cl$ eq | 12.1 | - |
| Respiratory Inorganics | RI | kg PM2.5 eq | 0.320 | - |
| Ionizing Radiation | IR | Bq C14 eq | $2.30 \times 10^3$ | - |
| Ozone Layer Depletion | OLD | kg CFC11 eq | $35.1 \times 10^{-6}$ | $483 \times 10^{-10}$ |
| Respiratory Organics | RO | kg $C_2H_4$ eq | 0.175 | - |
| Photochemical Oxidant | PO [g] | kg NMVOC | - | 5.26 |
| Human Toxicity | HT [h] | kg 1,4-DB eq. | - | 20.8 |
| Aquatic Ecotoxicity | AE [i] | kg TEG water | $256 \times 10^3$ | - |
| Freshwater Aquatic Ecotoxicity | FAE [h] | kg 1,4-DB eq. | - | 0.461 |
| Terrestrial Ecotoxicity | TE [i] | kg TEG soil | $57.4 \times 10^3$ | - |
| Terrestrial Acidification/Nutrification | TAN | kg $SO_2$ eq | 6.56 | 10.3 |
| Land Occupation | LO | $m^2$ organic arable | 8.05 | - |
| Aquatic Acidification | AA | kg $SO_2$ eq | 1.34 | - |
| Aquatic Eutrophication | AEU | kg $PO_4$ P-lim | $40.3 \times 10^{-3}$ | $5.25 \times 10^{-3}$ |
| Global Warming | GW | kg $CO_2$ eq | 398 | 1740 |
| Non-Renewable Energy | NRE | MJ primary | 6630 | - |
| Fossil Depletion | | kg oil eq. | - | 578 |
| Mineral Extraction | ME | MJ surplus | 6.55 | - |

## 6. Industrial Applications

There is a wide spectrum of potential industrial application for basalt FRC materials ranging from corrosion resistance equipment in chemical industry [63], wear and friction resistance equipments in automobile sector [151], principal parts for low velocity impact [152], to construction industry [153], energy industry [9], high temperature insulation for automobile catalysts [28], electrical appliances [9,28], fire hazard protection [144,154], and sports [9] equipment.

### 6.1. Automobile Industry

In the automotive sector, basalt fibre reinforced composites are widely used to manufacture car headliners. The superior recycling characteristics of basalt fibres provides a significant advantage for basalt composites to be used for production of car headliners. Composite laminates for car headliners constitute of a core with an adhesive layer provided adjacent opposing sides. Basalt FRC materials are provided adjacent to each adhesive layer [9,151]. Additionally, basalt fibre composites are used in automobile brake disk pads and clutch facings. Utilization of basalt FRC materials in these applications provide significant advantages as compared to using composites manufactured by other fibres like glass

which includes, better durability imparting longer service life period, superior wear and friction resistance, better shock resistance, higher working temperature, better chemical and moisture resistance, and additionally claimed to be eco-friendly [9,151].

### 6.2. Construction Industry

In the construction industry, basalt fibres are also widely used for several applications. Most common application of basalt fibres in construction industry is to produce basalt composite rebar for concrete reinforcement which provides many advantages as compared to steel and glass fibre rebar. Main advantages of basalt rebars includes superior fracture toughness than steel, lightweight, better chemical and moisture resistance, good thermal stability, ease in processability, good electrical conductivity [153]. Additionally, due to its superior corrosion resistance basalt rebars also provide good alternative to conventional materials for construction in marine sector and chemical plants [153].

Fibre reinforced concrete is widely used in the construction industry for its superior mechanical properties, such as flexural and tensile strength, good impact resistance, moisture resistance, better fracture toughness, and shock resistance. Basalt fibre reinforced concrete for construction industry provides a potential alternative to polypropylene and polyacrylonitrile fibre reinforced concrete owing to its better volumetric stability, excellent thermal resistance, good crack and impact resistance as well as being cost-effective [7,153].

Construction panels used for partitioning of building interior into rooms, elevator shafts, hallways require excellent fire resistivity and to work at high temperature. Basalt fibres imparting excellent fire resistance and thermal stability can act as a potential reinforcement to manufacture these panels [153].

### 6.3. Road Engineering

Short basalt fibre reinforcement in asphalt concrete can play a significant role in improving the performance of pavements imparting superior tensile strength, fracture toughness, deformation resistance [7,9,153].

### 6.4. Energy Industry

Basalt FRC materials can act as a potential alternative to E-glass FRC materials, which are widely used to manufacture different components in wind turbine blades, for example main spar, wing shell sections [7–9]. Carbon fibre-based composites are used alongside glass fibre reinforced composites to manufacture hybrid laminates for spar cap section of the main box spar in large wind turbine blades to account for the maximum bending stiffness of the blade. This helps the manufacturers to produce longer and lighter blades for maximum energy generation. However, application of basalt FRC with superior material properties can significantly improve the performance as well as recyclability and eco-friendliness of the blades. This can provide wind turbine blade manufacturers a significant improvement in cost-to-quality ratio [7–9]. For offshore wind energy instalments, steel based floating structures (tower and platform) are used which are prone to corrosion and moisture degradation. Basalt FRPs with good mechanical properties, superior moisture and corrosion resistance has the potential to replace steel floating structures for offshore wind turbine installations.

### 6.5. Sports Equipments

Basalt fibre reinforced polymers are widely used in sports equipment. Owing to it's superior moisture durability, corrosion resistance basalt FRC materials are used in water sports equipments such as kayaks, canoes, paddles, water skies. Other sports equipments enduring wide applications of basalt FRC materials are bicycle outer-ring parts, tennis rackets, skis, snowboards [154,155].

### 6.6. Other Applications

Basalt FRC materials can also be a good alternative to S-glass FRC materials used for ballistic applications. In the petrochemical industry, for production of pipes and compressed gas cylinders by filament winding technique [156,157], using basalt FRC materials with superior chemical resistance and thermal stability can help the manufacturers to improve cost-to-quality ratio of the product. Basalt fibres demonstrate excellent wet-out during fabrication process giving the filament winders a good opportunity to reduce the manufacturing cost as well as producing components with better material properties.

In power industry, carbon FRC based cores have been developed to replace metal cores for power transmission wires in high-voltage distribution lines. Basalt FRC with thermal conductivity and working temperature has the potential to replace carbon FRC cores in power transmission lines. Glass FRC are used in composite cross-arms to replace steel cross-arms for power transmission lines. Basalt FRC with superior insulation performance, excellent corrosion resistance, high strengths can replace glass FRC in composite cross-arms. Power distribution poles are generally based on traditional materials such as wood, metal, cement. Due to superior mechanical and chemical properties, composite poles based on glass FRC are being developed to replace traditional material poles. Basalt FRC with better mechanical properties, corrosion and moisture resistance, good thermal stability and electrical conductivity can be used a potential alternative to glass FRC in composite poles. In aviation industry, basalt FRC materials can be used for heat insulation purposes in the engine and fuselage sections [158]. In train industry, basalt FRC materials can be used for electro-insulation, heat and sound insulation of railway carriages [158].

### 7. Conclusions

A review of state of the art on basalt fibres and basalt fibre reinforced composites (FRC) have been presented in this paper. Common industrial applications of basalt fibre composites have also been discussed.

Basalt fibres procured directly from basalt rocks, is a potential sustainable cost-effective alternative to conventional fibres used in industrial composites. Basalt fibres demonstrate better mechanical properties, high thermal stability, good chemical and moisture resistance, sound insulation, good processability and is recyclable. In the reviewed literature, basalt FRC materials demonstrated good mechanical properties, thermal stability and moisture resistance compared to glass FRC materials, highlighting good fibre/matrix interfacial strength and compatibility with wide range of polymeric resin matrix.

Utilization of basalt FRC materials in different industrial applications provides a good cost-to-quality ratio for the manufacturers which can help them to significantly reduce the manufacturing cost without compromising the material mechanical properties.

This paper has highlighted significant advances made in the field of research based on basalt FRC materials, and wide spectrum of industrial application of the material. However, for effective adoption of basalt FRC materials in industrial applications, existing design guidelines and technical codes need to be optimized for design and development of new or existing structural components.

Following the literature review it is observed that previous studies in the literature agree regarding the potential of basalt fibre reinforced composites as an effective reinforcement for structural composites in industrial applications. However, for wider adoption of basalt fibres as the reinforcement for structural composites will require increase in mass production to meet the demand of the fibres, which can also lead to them becoming cost-effective with traditional reinforcements for structural composites, such E-glass.

**Author Contributions:** Conceptualization, I.R.C.; Methodology, I.R.C.; Writing—original draft preparation, I.R.C.; Writing—review and editing, I.R.C., R.P. and J.S. All authors have read and agreed to the published version of the manuscript.

**Funding:** This research received no external funding.

**Institutional Review Board Statement:** Not applicable.

**Informed Consent Statement:** Not applicable.

**Data Availability Statement:** Not applicable.

**Conflicts of Interest:** The authors declare no conflict of interest.

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
