# Peer review of "Developments and Industrial Applications of Basalt Fibre Reinforced Composite Materials"

_jcs, doi:10.3390/jcs6120367_

Round 1

Reviewer 1 Report

Dear Authors, indeed, Basalt FRC haven't penetrate Industrial area of applications which means somewhere is necessary to identify the reason for that. Probably because of production areas which is limited and now even more which makes things even harder! A Short review which tried to identify some selected areas which covered in most of cases after the review of 2015 at the same scientific area. Please find next some minor comments:

1. section 2 Basalt. page 3 line 116. Even less after war of Ukraine with Russia. I am not sure about the future of Basalt after that!

2. page 5. line 165 table 1. Please consider the prices after the war which will increase.

3. section 3. mechanical properties of basalt FRC. What about impact tests? low or high? Any input/data?

4. section 3. mechanical properties of basalt FRC. Any kind of hybrid using Basalt and Carbon or glass or Aramid or other natural fibers?

5. Please add if any new data about fire resistance, using neat or hybrid basalt fibers.

6. Please add references at section 6.3, 6.4 and 6.5 for applications which you have mentioned.

7. What about train or aeronautics/aerospace industry applications? 

Author Response

Please find the responses in the attachement

Reviewer 2 Report

The paper offers a review of state of the art on basalt fibres and basalt fibres reinforced composites. The topics are well balanced and the exposition is clear and concise. Only some typing or printing errors are to be reported.

- Page 1, line 32. Check the reference numbers inside the brakets: [1, 3] or [1, 4]?

- Page 8, line 279. The meaning of the ILSS acronim should be made explicit.

- Figure 3. Legend number 3. Check the reference number. In my opinion, it should be [87], not [88].

- Table 6. OLD row. Check the values reported in the last two columns. Is it possible that they are reversed?

- Page 20, line 688. Correct processibility in processability. 

Furthermore, I have one question to ask the authors concerning thermal stability of basalt fibre reinforced polypropylene composite. The authors report the results of a study (Tang et al., 2019) where the measured LOI of basalt fibre (BF) reinforced polypropylene is lower than LOI of pure polypropylene. Additionaly, the authors write that BF do not increase the initial decomposition temperature of polypropylene. So how is it possible that BF reinforced polypropylene composite demonstrate higher thermal stability than pure polypropylene? It is necessary to clarify the matter better. 

Author Response

Please find the responses in the attachment

Round 2

Reviewer 1 Report

Comments/replies accepted.

Reviewer 2 Report

No comment